# Clinical Competency in Managing Malnutrition–Sarcopenia Syndrome Among Physical Therapists: A Cross-Sectional Survey

**DOI:** 10.3390/nu17020281

**Published:** 2025-01-14

**Authors:** Roy Netzer, Netanel Levi, Kayla Ganchrow, Yfat Naan, Michal Elboim-Gabyzon

**Affiliations:** 1Department of Physical Therapy, Faculty of Social Welfare and Health Sciences, University of Haifa, Haifa 3498838, Israel; michal.elboim@gmail.com; 2Department of Physical Therapy, Shoham Medical Center, Pardes Hanna-Karkur 3701001, Israel; netanell@shoham.health.gov.il (N.L.);; 3Department of Diet and Nutrition, Shoham Medical Center, Pardes Hanna-Karkur 3701001, Israel

**Keywords:** malnutrition–sarcopenia syndrome, physical therapists, clinical competency, knowledge, attitude and beliefs, screening and treatment skills, interprofessional collaboration

## Abstract

Background/objectives: Malnutrition and sarcopenia are interrelated health concerns among the elderly. Each condition is associated with increased mortality, morbidity, rehospitalization rates, longer hospital stays, higher healthcare costs, and reduced quality of life. Their combination leads to the development of “Malnutrition–Sarcopenia Syndrome” (MSS), characterized by reductions in body weight, muscle mass, strength, and physical function. Despite being preventable and reversible through nutritional and physical interventions, the clinical competence of physical therapists (PTs) in managing MSS remains underexplored. This study aims to assess the clinical competency of PTs in MSS management. Methods: An anonymous cross-sectional survey was conducted from January to August 2024 among 337 certified PTs in Israel, using the “Qualtrics” platform. The survey assessed prior familiarity with MSS concepts, MSS knowledge levels, screening and treatment skills, attitudes and beliefs toward MSS management, and interprofessional collaboration practices. Results: While 52% of PTs were familiar with MSS, familiarity with diagnostic guidelines was low (EWGSOP2: 3.6%; GLIM: 0.6%). The MSS knowledge score was moderate, but screening and treatment skills were low. Attitudes toward MSS management were moderately positive, but self-belief in diagnosing and treating MSS was low. Interprofessional collaboration was limited, particularly in malnutrition care. PTs familiar with MSS had higher knowledge, better skills, more positive attitudes and beliefs, and greater interprofessional collaboration. Conclusions: Significant gaps exist in the clinical competency of Israeli PTs in MSS management. Integrating MSS content into physical therapy curricula and providing continuing professional development are necessary to enhance competencies. Equipping PTs with essential tools, clarifying roles, and promoting interprofessional collaboration can optimize MSS management and improve patient outcomes.

## 1. Introduction

The synergy between malnutrition and sarcopenia results in Malnutrition–Sarcopenia Syndrome (MSS) [1]. This clinical condition is characterized by significant reductions in body weight, muscle mass, strength, and physical function [1]. Specifically, it is defined by weight loss exceeding 5% within the past six months or more than 10% over a longer period; an Appendicular Skeletal Muscle Mass (ASM) below 20 kg for men and 15 kg for women, or an ASM/height^2^ below 7.0 kg/m^2^ for men and 5.5 kg/m^2^ for women, or a moderately and severely low Calf Circumference (CC) of 34 cm and 32 cm (males), and 33 cm and 31 cm (females), respectively; hand grip strength below 16 kg for women and 27 kg for men; and walking speed under 0.8 m/s in severe cases consistent with the European Working Group on Sarcopenia in Older People 2 (EWGSOP2) guidelines and the Global Leadership Initiative on Malnutrition (GLIM) criteria [2,3,4]. Malnutrition and sarcopenia are significant, interrelated health concerns among the elderly. Each condition is associated with advancing age and leads to increased morbidity, mortality, decreased quality of life, rehospitalization, prolonged hospital stays, and higher healthcare costs [1,5]. When sarcopenia and malnutrition co-occur, their impact on the individual is significantly magnified. Chen et al. (2022) analyzed the National Health and Nutrition Examination Survey (NHANES) cohort, including 12,469 participants aged 20 and above, and found that although being a minority (*n* = 239), individuals with both conditions faced nearly four times the risk of cardiovascular mortality and almost three times the risk of all-cause mortality compared to those with only one condition or healthy individuals [6]. This highlights the urgent need to integrate nutritional assessment with sarcopenia screening to optimize patient outcomes.

Recent studies have highlighted the overlap between sarcopenia and malnutrition. Ligthart-Melis et.al. (2020) found that 37% of hospitalized elderly patients had sarcopenia, and 66% were either malnourished or at risk of malnutrition, with an overlap of 41.6%, for sarcopenia in the presence of malnutrition (OR = 4.06, 95% CI: 2.43, 6.80; *p* < 0.001) [7]. Verstraeten et al. (2021) reported that 13% of geriatric patients had both conditions, with malnutrition significantly increasing the odds of severe sarcopenia (OR = 2.07, 95% CI = 1.13–3.81, *p* = 0.019) [8]. In Singapore, 35% of community-dwelling adults at nutritional risk were also sarcopenic [9]. Both malnutrition and sarcopenia are reversible and preventable conditions [10,11,12]. While no drug has yet been approved for these conditions, nutrition and physical activity, particularly personalized progressive resistance training with increased calories and protein intake, are the cornerstones in treating and preventing both conditions [10,11]. However, early identification is crucial in preventing rapid health deterioration [13].

The American Physical Therapy Association (APTA) emphasizes the role of physical therapists (PTs) in screening for nutritional issues, providing nutritional recommendations, and collaborating with dietetic professionals to optimize patient outcomes [14]. Despite their role in managing MSS, the clinical competence of PTs in this area remains under-explored. Clinical competence in healthcare includes behaviors indicating exceptional performance, such as clinical skills, interpersonal skills, problem-solving, judgment, and technical expertise [15]. This competence is vital for setting standards, aligning with evidence-based care, and fostering collaboration, particularly between physical therapy and nutrition professions [15,16]. Such collaboration is crucial for effectively managing malnutrition and sarcopenia [5,16]. Studies exploring PTs competencies in addressing these conditions are limited, with most focusing on each condition separately [17,18,19,20,21,22]. Research indicates that while most PTs recognize their responsibility for managing sarcopenia, many lack the requisite skills to diagnose and treat it in accordance with established guidelines. Consequently, the practical implementation of sarcopenia diagnosis and treatment by PTs remains insufficient [17,19,20,22]. Similarly, although some studies highlight PTs’ willingness to incorporate dietary assessment and counseling into their practice [23,24], they often lack confidence and do not utilize established criteria when screening and treating malnutrition [18,22]. Existing studies, however, tend to examine malnutrition and sarcopenia separately rather than as an integrated condition. Moreover, existing research often combines PTs with other healthcare professionals, obscuring the unique skills and contributions of PTs in MSS care. To address this gap, it is crucial to independently examine the distinct competencies of PTs in managing MSS. Building on our prior work [23,25], this study focuses on clinical competency in MSS management, introducing novel assessment tools and highlighting the importance of interprofessional collaboration. Specifically, it aims to assess their competence in managing MSS in terms of PTs familiarity with MSS concepts; their knowledge, clinical skills, attitudes and beliefs toward MSS management; and their collaboration with other healthcare professionals in MSS care

## 2. Methods

### 2.1. Study Design

An anonymous nationwide online self-report survey using “Qualtrics” software [February 2024] (Qualtrics, Provo, UT, USA) was conducted between 11 January 2024 and 11 August 2024. This study was approved by the Ethics Committee at the Faculty of Social and Health Sciences, University of Haifa (Approval number 141/23). Completion and submission of the questionnaire indicated informed consent, as per the accompanying online instructions.

### 2.2. Participants

#### Eligibility and Exclusion Criteria

Certified PTs with a bachelor’s degree and at least one year of clinical experience were eligible to participate. PTs with an additional degree as registered dietitians were excluded.

### 2.3. Study Procedures

#### Survey Instrument and Outcome Measures

A questionnaire was developed and adapted for Israeli PTs, incorporating newly formulated questions alongside translated items from four existing English-language questionnaires previously used in studies on sarcopenia and malnutrition [17,19,20,26]. The newly formulated questions were based on recommendations from international working groups [2,3] and recent studies on malnutrition and sarcopenia syndrome [1,27,28,29].

The questionnaire comprised six main sections:Demographic details: Personal and professional information, including age, gender, academic degrees, years of clinical experience, place of work, and the age range of their patients.Prior familiarity with malnutrition, sarcopenia, and MSS concepts: This section assessed participants’ prior familiarity through three yes/no questions asking whether they had previously heard of the EWGSOP2 criteria, the GLIM criteria, and MSS. Participants who indicated familiarity with MSS were further asked where they first learned about it. For detailed responses, please refer to Table A1.Knowledge level regarding malnutrition, sarcopenia, and MSS: This outcome included 16 questions assessing knowledge in three domains: sarcopenia (5 questions), malnutrition (7 questions), and MSS (4 questions). The questions were presented in multiple-choice format (e.g., “Malnutrition refers primarily to a deficiency of what?” with response options as follows: (a) I don’t know. (b) Proteins and caloric intake. (c) Fats and proteins. (d) Carbohydrates and vitamins. (e) None of the above) or true/false/don’t know format (e.g., “Sarcopenia is part of the natural aging process”). To provide a comprehensive evaluation, we assessed five sub-knowledge areas: (1) sarcopenia knowledge, (2) malnutrition knowledge, (3) MSS-specific knowledge, (4) combined sarcopenia and malnutrition knowledge, and (5) overall MSS knowledge. Each of these scores is calculated by summing the responses for all relevant questions and dividing by the total number of questions within each category.Assessment of sarcopenia and malnutrition screening and treatment skills: This outcome measure consisted of five multiple-choice questions evaluating participants’ skills in MSS screening and treatment (e.g., “Which indicator(s) suggest a patient is ’at risk’ for malnutrition?” with response options (a) I don’t know. (b) Abnormal levels of B12 and folic acid. (c) Lack of strength and fatigue. (d) Unplanned weight loss. e. BMI below 18.5 kg/m^2^”). Scores were calculated by summing the correct responses and dividing by the total number of questions (out of 100), with higher scores indicating greater proficiency. For detailed responses, please refer to Table A2.Evaluation of attitudes and beliefs toward managing MSS in clinical practice: This outcome measure included six statements rated on a five-point Likert scale from 1 (strongly disagree) to 5 (strongly agree), assessing attitudes (e.g., “Assessing the risk of malnutrition is an essential part of physical therapy treatment”) and beliefs (e.g., “I am confident that I have sufficient knowledge to manage sarcopenia”) toward MSS management. The total score was calculated by summing the responses to all six statements, resulting in a possible range from 6 to 30, with higher scores indicating more favorable attitudes and beliefs.Evaluation of interprofessional collaboration in MSS care: This section assessed collaboration with other healthcare professionals in managing sarcopenia and malnutrition:
-Sarcopenia Interprofessional Collaboration-Participants were asked to report on whether they collaborated with other professionals in the management of sarcopenia. The question posed was as follows: “Which healthcare professionals do you consult for sarcopenia management?” The following response options were given: (a) I do not diagnose or treat sarcopenia; (b) I manage sarcopenia independently; (c) I consult with doctors; (d) I consult with nurses; (e) I consult with another PTs; (f) I consult with a fitness trainer/exercise physiologist; and (h) I consult with a clinical dietitian. For analysis, responses (c) through (h) were grouped to represent collaboration with other healthcare professionals.-Malnutrition Interprofessional Collaboration-Participants rated the level of collaboration with other health professionals in diagnosing and treating malnutrition within their workplace. This was measured on a four-point Likert scale, ranging from 1 (no collaboration) to 4 (very good collaboration), with higher scores reflecting greater levels of interprofessional collaboration.


### 2.4. Expert Panel Review

The questionnaire underwent linguistic editing and was evaluated for content validity and clarity by an expert panel, consisting of three PTs and two clinical nutritionists. Additionally, two focus groups of PTs from a general hospital and a rehabilitation center reviewed the questionnaire. Feedback from these focus groups resulted in several revisions, including the addition of a question regarding the age range of participants’ patients, specific inquiries about “home visits”, the inclusion of “don’t know” and “I didn’t think of it” response options, and a reduction in the length of the questionnaire from 82 to 60 questions.

### 2.5. Data Collection Procedures

The revised questionnaire was implemented as an anonymous survey using “Qualtrics” software platform [February 2024] (Qualtrics, Provo, UT, USA). The survey was distributed to licensed qualified PTs in Israel through convenience sampling. Recruitment advertisements were posted on social media platforms, including Facebook and WhatsApp groups dedicated to physical therapy groups, as well as through the Association for the Advancement of Physical therapy in Israel, and the Physical therapy Department Secretariat. The survey was also shared among clinical instructors, PTs pursuing advanced degrees, and personal contacts.

### 2.6. Statistical Analysis

Descriptive statistics were calculated for all outcome measures. The internal consistency of the five knowledge-level sections was determined using bivariate Spearman correlations, with correlations categorized as weak (r ≤ 0.3), moderate (0.3 < r ≤ 0.6), and strong (r ≥ 0.7) [30]. Spearman’s correlation was chosen because the knowledge test comprises multiple-choice questions with dichotomous correct/incorrect answers, resulting in ordinal data that may not be normally distributed [30].

The internal consistency of the attitudes and beliefs scale was determined using Cronbach’s alpha. This approach is appropriate for Likert-scale items that produce continuous data with consistent directionality [31]. Alpha values were classified as follows: excellent (α ≥ 0.9), good (0.7 ≤ α < 0.9), acceptable (0.6 ≤ α < 0.7), poor (0.5 ≤ α < 0.6), and unacceptable (α < 0.5) [31].

Bivariate Spearman correlation coefficient tests were utilized to assess relationships between all continuous outcome variables, including the five MSS knowledge test sections scores, assessment and treatment skills score, attitudes and beliefs score, and malnutrition interprofessional collaboration scores.

Differences in outcome variables were analyzed using the Wilcoxon two-sample test. Specifically, comparisons were made between genders, between participants who reported engaging in interprofessional collaboration for sarcopenia management and those who did not, and between participants with and without prior familiarity with the MSS concept. Statistical significance was set at *p* < 0.05.

### 2.7. Results

#### 2.7.1. Demographic and Professional Information

A total of 337 PTs participated in the survey. The participants’ mean age was 40.9 ± 10.6 years, with an average of 14.1 ± 11.4 years of professional experience. A majority were female (72.4%) and had completed their education in Israel (97%). A significant portion (43.6%) worked in public outpatient clinics, and 52.8% conducted home visits in addition to their primary workplace responsibilities. Most participants (82.3%) reported that a majority of their patients were adults over the age of 18 (Table 1).

#### 2.7.2. Knowledge Level Regarding Malnutrition, Sarcopenia, and MSS

More than half of the participants (52.5%) reported being familiar with the concept of MSS. However, familiarity with specific diagnostic guidelines was notably low, with only 3.6% familiar with the EWGSOP2 guidelines and just 0.6% familiar with the GLIM criteria. The primary sources of knowledge were undergraduate studies (45.6%) and reading academic articles (23.1%).

Bivariate correlation analysis revealed significant correlations between the five knowledge-level sections, indicating strong internal consistency (r = 0.66–0.88, *p* < 0.001) (Table 2).

The overall MSS knowledge test score averaged 69.6 ± 13.3 out of 100, reflecting moderate knowledge. Participants scored 66.8 ± 19.7 on sarcopenia-related questions and 71.4 ± 14.7 on malnutrition-related. It was found that 91% of PTs knew that sarcopenia is treatable. However, 59.5% did not know the definition of sarcopenia. While 70% correctly believed that malnutrition primarily involves protein and calorie deficiencies, only 6.5% could identify its most common clinical signs. The MSS-specific knowledge score averaged 69.9 ± 25.1, with 75% of participants acknowledging recognizing the co-occurrence of malnutrition and sarcopenia; however, only 26.4% knew the most affected populations, for more details see Table 3.

Participants with prior familiarity of MSS scored significantly higher in the sarcopenia knowledge score (69.6 ± 18.5), MSS-specific knowledge section (77.0 ± 20.8), the combined malnutrition and sarcopenia knowledge (71.4 ± 67.7), and total overall MSS knowledge score (72.8 ± 11.6) compared to those without prior familiarity (64.2 ± 20.4, 63.4 ± 26.9, 67.7 ± 13.6, 66.7 ± 13.9, respectively; *p* < 0.001). All knowledge scores were not significantly influenced by gender (Table A3). 

#### 2.7.3. Assessment of Sarcopenia and Malnutrition Screening and Treatment Skills

The average skills score was 30.7 ± 19.9 out of 100, indicating an overall generally low proficiency in sarcopenia and malnutrition screening and treatment skills. Screening skills were notably deficient: only 6.2% were familiar with clinical diagnostic methods for sarcopenia, and 22.9% could identify key indicators of malnutrition risk. While 44.2% were familiar with sarcopenia treatment strategies, only 30.6% could specify the most appropriate nutritional recommendation for malnutrition management (Table A2). 

A weak but statistically significant correlation was found between the skill score and the overall MSS knowledge score (r = 0.16, *p* < 0.01), as well as the combined sarcopenia and malnutrition knowledge score (r = 0.13, *p* < 0.05) (Table A4). 

Participants who were familiar with MSS prior to the study demonstrated higher skills in screening and treatment (33.0 ± 20.2) compared to those without prior familiarity (28.7 ± 19.5; *p* = 0.05). No significant differences in skill scores were found between genders (Table A5). 

#### 2.7.4. Attitudes and Beliefs Toward Managing MSS in Clinical Practice

The attitudes and beliefs scale demonstrated good internal consistency, with a Cronbach’s alpha of 0.7. The average score was 18.0 ± 3.4 in a scale of 6–30, indicating a moderate level of attitudes and beliefs toward managing MSS. Participants highly agreed on the importance of diagnosing and treating sarcopenia (4.5 ± 0.6 out of 5) and malnutrition (4.2 ± 0.7 out of 5) within the scope of physical therapy practice. However, the self- belief in their ability to diagnose and treat these conditions was noticeably low (2.4 ± 1.0 and 2.1 ± 1.0 out of 5, respectively) (Table A6). 

Regarding the attitudes for perceived responsibility, medical doctors were most frequently identified as responsible for diagnosing and treating sarcopenia (50.5%), followed by PTs (35.6%). For malnutrition management, registered dietitians (70%) and medical doctors (63.5%) were seen as primarily responsible, with only 9.2% identifying PTs as responsible. Significant positive correlations were observed between the attitudes and beliefs scores and various knowledge and skill scores, including the combined sarcopenia and malnutrition knowledge score (r = 0.28, *p* < 0.001) and the overall MSS knowledge score (r = 0.28, *p* < 0.001) (Table 4). Participants who were familiar with MSS exhibited more positive attitudes and beliefs (18.8 ± 3.5) compared to those without prior familiarity (17.2 ± 3.1; *p* < 0.001). Gender did not significantly influence the attitudes and beliefs (Table A7). 

### 2.8. Interprofessional Collaboration in MSS Care

Table 5 provides participants’ behaviors in MSS care, focusing on the extent of interprofessional collaboration. Many participants (59.4%) reported collaborating with healthcare professionals in the management of sarcopenia, primarily with clinical dietitians (44.2%), medical doctors (36.7%), and other PTs (28.9%). In contrast, the interprofessional collaboration score for malnutrition was moderate, with an average of 2.13 ± 1.0 out of 4. Notably, 65% of PTs indicated little-to-no cooperation with other healthcare professionals in malnutrition care. The primary reasons cited for the limited collaboration were that 36.5% of participants felt healthcare professionals lack awareness of the importance of interprofessional collaboration, and 33% believed that diagnosing and treating malnutrition is solely the responsibility of dietitians.

Participants who were familiar with MSS demonstrated significantly higher malnutrition interprofessional collaboration scores (2.3 ± 1.0) compared to those without prior awareness (2.0 ± 1.0; *p* < 0.01) (Table A8). Similarly, participants with MSS familiarity were more likely to collaborate in sarcopenia treatment (58%) (Table A9). Gender did not significantly influence these outcomes. Significant correlations were found between the malnutrition collaboration score and knowledge metrics, including the malnutrition knowledge score (r = 0.12, *p* < 0.05) and the combined sarcopenia and malnutrition knowledge score (r = 0.15, *p* < 0.01). Additionally, a correlation was observed between the malnutrition collaboration score and the attitudes and beliefs score (r = 0.22, *p* < 0.001), indicating that more positive attitudes and greater self-beliefs in managing MSS are linked to enhanced collaborative approaches (Table A10).

Regarding sarcopenia interprofessional collaboration, significant differences were found in all five sections of the knowledge test scores, attitudes and beliefs scores, and clinical skills scores between professionals who engage in cooperation and those who do not (Table 6). These results suggest that PTs who collaborate with other healthcare professionals in sarcopenia management possess higher levels of MSS knowledge, more positive attitudes and beliefs toward MSS care, and greater screening and treatment skills compared to those who do not collaborate.

### 2.9. Discussion

This study uniquely examined PTs’ knowledge of MSS, addressing malnutrition and sarcopenia as interconnected health issues in the elderly. A moderate overall MSS knowledge score of 69.6% was found, with only 45.6% of participants exposed to MSS topics during undergraduate studies and 11.3% during postgraduate studies. Given the absence of prior studies specifically focusing on PTs’ knowledge of MSS, direct comparisons are not available.

The current findings reveal moderate knowledge levels among PTs regarding sarcopenia, with only 41.5% of participants correctly identifying its definition (low muscle mass, strength, and physical function). This highlights a significant knowledge gap among the surveyed PTs. These results are notably lower compared to those reported in previous studies, where 76.8–96% of PTs correctly identified the definition of sarcopenia [17,19]. These discrepancies may stem from differences in study populations and assessment tools, and the absence of a universally standardized definition of sarcopenia [32]. This discrepancy underscores the need for targeted educational initiatives to enhance PTs’ understanding of sarcopenia. Similarly, the current results revealed moderate knowledge levels among PTs regarding malnutrition, aligning with findings from previous studies [25,33,34]. These results underscore the necessity of integrating comprehensive nutrition education into physical therapy curricula, as endorsed by the Commission on Accreditation in Physical Therapy Education (CAPTE) and supported by PTs [23,35,36].

Assessment of PTs’ clinical skills in MSS management revealed significant deficiencies, with an average score of 30.7/100. Screening skills were especially low: only 6.2% were familiar with sarcopenia diagnostic methods, and 22.9% could identify malnutrition risk indicators. Familiarity with diagnostic guidelines was minimal, with just 3.6% aware of EWGSOP2 and 0.6% of GLIM criteria. Treatment skills were also lacking, with just 45% of PTs understanding that sarcopenia treatment requires both physical and nutritional interventions, with 50% unsure of appropriate actions for suspected malnutrition. Our findings, particularly the uncertainty among PTs regarding appropriate actions for suspected malnutrition and sarcopenia management, reveal a significant gap in both education and clinical practice. Similar studies have reported limited skills and familiarity with guidelines, highlighting a widespread issue [17,18]. This emphasizes the need to integrate foundational knowledge of malnutrition and sarcopenia into PT curricula, alongside practical guidelines for screening and interdisciplinary referral. These initiatives would enhance PTs’ ability to manage MSS effectively within a multidisciplinary team. Addressing these gaps is essential, as they contribute to underdiagnosis and poor patient outcomes, with evidence showing that familiarity with guidelines improves clinical skills [2,3].

This study is the first to assess PTs attitudes and beliefs regarding MSS management. While PTs acknowledged the importance of diagnosing and treating sarcopenia (4.5/5) and malnutrition (4.2/5), their self-belief in these abilities was low (2.4 and 2.1/5, respectively). This gap aligns with previous studies which found that PTs recognized nutritional or sarcopenia management’s significance but lacked confidence to implement it [17,23]. Significant positive correlations between PTs’ attitudes and beliefs, knowledge, and clinical skills suggest that greater MSS knowledge and clinical skills enhance attitudes and beliefs toward MSS care. Notably, PTs with prior MSS familiarity showed more positive attitudes, underscoring the impact of targeted education on promoting evidence-based MSS care. This coincides with previous studies that associate prior nutritional education with more favorable attitudes toward integrating nutrition into physical therapy practice [24,37]. Future studies should explore whether positive attitudes and perceived competence drive MSS integration.

A significant portion of PTs reported limited collaboration with other healthcare professionals, particularly in managing malnutrition. In total, 65% of PTs indicated minimal-to-no cooperation in malnutrition care, while 59.4% reported collaborating with others in sarcopenia management, primarily dietitians (44%). This disparity may reflect the delegation of malnutrition care to dietitians, as only 9.2% of PTs felt responsible for managing malnutrition. These findings align with Reijnierse et al. (2017), who found that 59.9% of PTs collaborated in sarcopenia management, mainly with dietitians (87%); and with Reinders et al. (2022), who highlighted limited collaboration in malnutrition care due to unclear role definitions [16,19].

Our results further demonstrate that higher levels of MSS-related knowledge, clinical skills, and positive attitudes and beliefs among PTs were associated with improved collaboration, particularly in sarcopenia management. However, malnutrition care showed weaker links to interprofessional collaboration, likely due to role ambiguity and insufficient training. This is consistent with previous studies identifying unclear roles and knowledge gaps as barriers to effective collaboration [16,19,35]. Mayerele et al. (2000) emphasized that teamwork with dietitians enhances PTs’ confidence and nutritional knowledge, underscoring the value of interprofessional approaches. [35] Although nutritional care is within the PTs’ professional scope, as recognized by the APTA, its implementation varies globally due to legal and systemic constraints [14,38]. The expected clinical competencies of PTs in Israel currently do not include specific provisions for nutritional screening. PT education in Israel provides comprehensive training in basic and applied sciences, clinical reasoning, and patient management. However, nutrition is not formally integrated into the core competencies. Some physical therapy programs offer elective courses in nutrition as part of undergraduate studies. Addressing these gaps requires a clear definition of PTs’ roles in nutritional management, along with the promotion of interprofessional collaboration through healthcare systems and professional organizations. Such efforts are essential for enhancing patient outcomes and reducing healthcare costs, particularly in the management of MSS.

This study has several limitations. The reliance on self-reported data may introduce biases such as overestimation or underestimation competencies. Additionally, the use of convenience sampling of Israeli PTs and survey distribution through social networks may limit generalizability. Furthermore, the study did not examine the potential relationship between age and clinical competency in MSS management, nor did it directly evaluate PTs’ awareness of patients’ current engagement with other healthcare professionals. Such limitations underscore the need for further research to better understand PTs’ clinical competency in MSS care.

## 3. Conclusions

This study highlights significant gaps in Israeli PTs’ clinical competency in MSS management. While PTs show moderate knowledge and positive attitudes toward MSS management, they lack the essential skills and self-belief needed for effective care. Interprofessional collaboration is limited, especially in malnutrition care, possibly due to role perception. PTs familiar with MSS had better knowledge, clinical skills, attitudes and beliefs, and interprofessional collaboration. This underscores the need to improve academic curricula and continuing education. Providing PTs with role clarity, essential tools, and collaboration opportunities can enhance MSS management and patient outcomes. Educational and healthcare institutions should prioritize these initiatives. Future research should assess how PT competencies affect MSS care and evaluate educational interventions for long-term effectiveness.

## Figures and Tables

**Table 1 nutrients-17-00281-t001:** Demographic and professional information (*n* = 337).

Characteristics		
Age, years, mean (SD)		40.9 (±10.6)
Gender, female, N (%)		244 (72.4%)
University, Israel, N (%)		327 (97%)
Professional experience, years, mean (SD)		14.1 (±11.4%)
Having additional degree, N (%)	1. Master’s degree	108 (32.1%)
	2. Doctorate	15 (4.5%)
	3. Bachelor’s degree (not in physical therapy)	20 (5.9%)
	4. No	194 (57.6%)
Workplace, N (%) **	1. Public outpatient clinics	147 (43.6%)
	2. Acute hospital	40 (11.9%)
	3. Private outpatient clinics	92 (27.3%)
	4. Rehabilitation hospitals	94 (27.9%)
	5. Academia	14 (4.2%)
	6. Child development	25 (7.4%)
	7. Sports field	28 (8.3%)
	8. Israel Defense Forces	7 (2.1%)
	9. Home Visits	29 (8.6%)
	10. Other	54 (16%)
Do you conduct home visits as an adjunct to your main workplace? N (%)	1. Yes	178 (52.8%)
	2. No	159 (47.2%)
What is the age range of your patients? N (%) **	1. <18 year	32 (9.5%)
	2. ≥18 years	107 (31.8%)
	3. ≥65 years	115 (34.1%)
	4. All age range	164 (48.7%)

** Participants could select multiple options.

**Table 2 nutrients-17-00281-t002:** Correlation results of the knowledge-test sections’ scores (*n* = 337).

	Malnutrition Knowledge Score	Malnutrition and Sarcopenia Knowledge Score	MSS Knowledge Score	Total Knowledge
Sarcopenia knowledge score	0.194 ^a^	0.762 ^a^	0.176 ^a^	0.664 ^a^
	<0.001	<0.001	0.001	<0.001
Malnutrition knowledge score		0.763 ^a^	0.222 ^a^	0.682 ^a^
		<0.001	<0.001	<0.001
Malnutrition and sarcopenia knowledge score	-	-	0.258 ^a^	0.880 ^a^
			<0.001	<0.001
MSS knowledge score	--	-	-	0.653 ^a^
				<0.001

^a^ Spearman correlation coefficients. MSS = Malnutrition–Sarcopenia Syndrome.

**Table 3 nutrients-17-00281-t003:** Malnutrition, sarcopenia, and MSS knowledge test results (*n* = 337).

**Question**	**Correct Answer, N (%)**
Knowledge of sarcopenia	
1. What is sarcopenia?	140 (41.5)
2. Sarcopenia is part of aging **	238 (70.6)
3. Sarcopenia is preventable **	219 (65.0)
4. Sarcopenia is treatable **	307 (91.1)
5. Obesity lowers sarcopenia risk **	221 (65.6)
Sarcopenia knowledge score, mean ± std, median [Q1, Q3]:sum of 5 items/100 (percent)	66.8 ± 19.7, 60.0 [60.0, 80.0]
knowledge of malnutrition	
6. Malnutrition refers to deficient mainly of what?	235 (69.7)
7. What is the most common clinical manifestation of malnutrition?	22 (6.5)
8. Which population groups are “at risk” of malnutrition in the community?	291 (86.4)
9. Malnutrition is part of aging **	292 (86.7)
10. Malnutrition is preventable **	323 (95.9)
11. Malnutrition is treatable **	330 (97.9)
12. Obesity lowers malnutrition risk **	192 (57.0)
Malnutrition knowledge score, mean ± std, median [Q1, Q3]: sum of 7 items/100 (percent)	71.4 ±14.7, 71.4 [57.1, 85.7]
Malnutrition and sarcopenia knowledge score, mean ± std, median [Q1, Q3]: sum of 12 items/100 (percent)	69.5 ±13.2, 75.0 [58.3, 75.0]
knowledge of the concept of MSS	
1. In which population groups MSS is more common?	89 (26.4)
2. The presence of malnutrition may be a predisposing factor for the development of sarcopenia in the elderly. **	303 (89.9)
3. The presence of malnutrition and sarcopenia together constitutes a prognostic index (predictive) for an increase in hospitalization time, re-hospitalizations, and mortality rate among hospitalized elderly. **	297 (88.1)
4. In many patient populations there are patients suffering from malnutrition and sarcopenia **	253 (75.1)
MSS Knowledge score, mean ± std, median [Q1, Q3]: sum of 4 items/100 (percent)	69.9 ± 25.1, 75.0 [50.0, 75.0]
Toal Knowledge Test score, mean ± std, median [Q1, Q3]: sum of 16 items/100 (percent)	69.6 ± 13.3, 75.0 [62.5, 81.3]

MSS = Malnutrition–Sarcopenia Syndrome. ** True/false/don’t know questions.

**Table 4 nutrients-17-00281-t004:** Bivariate correlation coefficient analysis between the attitudes and beliefs score and knowledge test sections and skills scores (*n* = 337).

Variable	Attitudes and Beliefs Score
	r (*p*-Value)
Sarcopenia and malnutrition screening and treatment skill score ^a^	0.15 (<0.01)
Sarcopenia knowledge score ^a^	0.24 (<0.001)
Malnutrition knowledge score ^a^	0.19 (<0.001)
MSS knowledge score ^a^	0.15 (<0.01)
Sarcopenia and malnutrition knowledge score ^a^	0.28 (<0.001)
Overall MSS knowledge score ^a^	0.28 (<0.001)

^a^ Spearman correlation coefficients. MSS = Malnutrition–Sarcopenia Syndrome.

**Table 5 nutrients-17-00281-t005:** Behavior in terms of interprofessional collaboration in MSS care.

Question		Answer
Which health professionals do you consult for the purpose of providing an intervention for sarcopenia? ** N (%) (29 missing)	1. I do not diagnose sarcopenia and therefore do not treat sarcopenia at the clinic either	115 (37.3)
	2. I take care of myself without consulting	10 (3.3)
	3. With doctors	113 (36.7)
	4. With nurses	20 (6.5)
	5. With another physical therapist/colleague	89 (28.9)
	6. With a fitness trainer/exercise physiologist	38 (12.3)
	7. With a clinical dietitian	136 (44.2)
	8. Other	10 (3.3)
If you answered that you treat sarcopenia without consulting other health professionals, what are the possible reasons for not being referred? ** N (%) (308 missing)	1. I see no need	2 (6.9)
	2. The absence of other health professionals in the workplace (for example, a private institute)	11 (37.9)
	3. Lack of cooperation in treatments between health professionals in my workplace.	13 (44.8)
	4. I don’t know which of the health professionals to turn to for advice on the matter	6 (20.7)
What is the level of cooperation between you and other health professionals (doctors, nurses, clinical dietitians) in the diagnosis and treatment of malnutrition at your workplace? N (%)	1. No cooperation	115 (34.1)
	2. Partial cooperation	103 (30.6)
	3. Good cooperation	80 (23.7)
	4. Very good cooperation	39 (11.6)
Malnutrition interprofessional collaboration Score, mean ± std, median [Q1, Q3]:		2.13 ± 1, 2 [1, 3]
If you answered that there is no cooperation, what do you think are the possible reasons for this situation? ** (222 missing)	1. Lack of knowledge of who to turn to	33 (28.7)
	2. Healthcare professionals lack awareness of the importance of interprofessional collaboration	42 (36.5)
	3. Absence of additional health professionals in the workplace	29 (25.2)
	4. Giving sole responsibility for diagnosis and treatment of the issue to nutritionists in my workplace	38 (33)
	5. Other	15 (13)

** Participants could select multiple options.

**Table 6 nutrients-17-00281-t006:** Comparative analysis of the sarcopenia interprofessional collaborations scores compared with the knowledge test sections, screening and treatment skill score and attitude and beliefs.

	Sarcopenia Interprofessional Collaborations	N	Mean	STD	Median	*p*-Value
Sarcopenia knowledge score ^a^	Non-collaborating	115	60.3	20.7	60.0	<0.001
	Collaborating	183	70.1	18.3	80.0	
Malnutrition knowledge score ^a^	Non-collaborating	115	67.8	15.9	71.4	0.007
	Collaborating	183	73.1	14.0	71.4	
MSS-specific knowledge score ^a^	Non-collaborating	115	64.7	14.4	66.7	<0.001
	Collaborating	183	71.9	11.7	75.0	
Combined malnutrition and sarcopenia knowledge score^ a^	Non-collaborating	115	63.9	29.5	75.0	0.007
	Collaborating	183	73.4	21.7	75.0	
Total knowledge test score^ a^	Non-collaborating	115	64.5	15.5	68.8	<0.001
	Collaborating	183	72.2	10.9	75.0	
Skill score^ a^	Non-collaborating	115	26.6	17.3	20.0	<0.005
	Collaborating	183	33.9	21.0	40.0	
Attitudes and beliefs score^ a^	Non-collaborating	115	16.0	2.5	16.0	<0.001
	Collaborating	183	19.1	3.3	19.0	

MSS = Malnutrition–Sarcopenia Syndrome. ^a^ Wilcoxon two-sample test.

## Data Availability

Data are available upon request due to restrictions, e.g., privacy or ethical.

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
