# Peer review of "Clinical Competency in Managing Malnutrition–Sarcopenia Syndrome Among Physical Therapists: A Cross-Sectional Survey"

_nutrients, 2025, doi:10.3390/nu17020281_

Round 1

Reviewer 1 Report

Comments and Suggestions for Authors

This study is interesting and important. However, the following points should be checked up before publication.

1. This study is similar to recent papers by the same authors (Reference No. 31-32. Netzer R, Elboim-Gabyzon M. 2023). Therefore, it is necessary to clearly explain the novelty,  originality, and impact of this study. 

2. Is the clinical competency of PTs in MSS management affected by age?

3. What is “IDF” in Table 1?

Reviewer 2 Report

Comments and Suggestions for Authors

Dear Authors, Thank you for your manuscript on clinical competency in managing malnutrition-sarcopenia syndrome among physical therapist: a cross sectional study survey.   These are my comments and suggestions:

Introduction:

Please provide a more precise definition of MSS - ?how much weight loss, muscle mass loss, and decrease in strength / physical function

It is mentioned about the role of PT in screening for nutritional issues as per the APTA - however this is a survey of PTs in Israel - please expand on how PTs are trained and certified in Israel?  Is the nutritional screening part of the clinical competencies for PTs in Israel as per their national body?

Results

It is mentioned that only 3.6% were familiar with EWGSOP2 guidelines and 0.6% familiar with GLIM criteria - is this knowledge part of the current clinical competencies for PT education in Israel?  Similarly, it is mentioned that 50% were unsure of appropriate actions for suspected malnutrition - is this part of their expected clinical competency?

Management of MSS Syndrome is complex and requires multi-disciplinary collaboration - did any of the survey question address knowledge of the patients' current involvement with other health professionals such as dieticians or family doctors / geriatric physicians ?   ie were the PTs aware if a dietician was already seeing their patient?  Please expand discussion about the expected scope of PTs in management of MSS and how interdisciplinary  referrals / collaboration may assist in the care of this complex entity.

Thanks again for your manuscript contribution.
